# Enhancing Fermentation Process Monitoring through Data-Driven Modeling and Synthetic Time Series Generation

**DOI:** 10.3390/bioengineering11080803

**Published:** 2024-08-08

**Authors:** Hyun J. Kwon, Joseph H. Shiu, Celina K. Yamakawa, Elmer C. Rivera

**Affiliations:** 1School of Engineering, Andrews University, Berrien Springs, MI 49104, USA; ccoparivera@andrews.edu; 2Department of Computer Science, Andrews University, Berrien Springs, MI 49104, USA; shiuj@andrews.edu; 3River Stone Biotech ApS, Fruebjergvej 3, 2100 Copenhagen, Denmark; celinayamakawa@gmail.com

**Keywords:** variational autoencoders, soft sensor, deep learning, modeling, fermentation processes

## Abstract

Soft sensors based on deep learning regression models are promising approaches to predict real-time fermentation process quality measurements. However, experimental datasets are generally sparse and may contain outliers or corrupted data. This leads to insufficient model prediction performance. Therefore, datasets with a fully distributed solution space are required that enable effective exploration during model training. In this study, the robustness and predictive capability of the underlying model of a soft sensor was improved by generating synthetic datasets for training. The monitoring of intensified ethanol fermentation is used as a case study. Variational autoencoders were employed to create synthetic datasets, which were then combined with original datasets (experimental) to train neural network regression models. These models were tested on original versus augmented datasets to assess prediction improvements. Using the augmented datasets, the soft sensor predictive capability improved by 34%, and variability was reduced by 82%, based on R^2^ scores. The proposed method offers significant time and cost savings for dataset generation for the deep learning modeling of ethanol fermentation and can be easily adapted to other fermentation processes. This work contributes to the advancement of soft sensor technology, providing practical solutions for enhancing reliability and robustness in large-scale production.

## 1. Introduction

Fermentation processes have become increasingly relevant to produce diverse products including fuels, chemicals, food, and pharmaceuticals. Modeling in fermentation processes is very important to better understand the metabolic pathways and their correlation with process parameters, and it can be used as a tool to identify the bottlenecks of the cell factories. Although we collect more data in many fermentation processes today, the domain of these processes does not fall into the big data category, limiting the applicability of data-driven modeling. Optimization, control, and techno-economic evaluation based on poorly calibrated models can yield unreliable results.

Previous work by Rivera et al. [1] has demonstrated that it is possible to predict the titer of the fermentation product using auxiliary variables using tools from process analytical technology that are far easier to measure. Specifically, electrical capacitance, redox potential, pH, and temperature were used as features in a feedforward neural network regression model to prototype a soft sensor. After imputing missing data as well as corrupt data from improperly calibrated sensors, they were able to achieve satisfactory preliminary results. However, in an industrial environment, sensor probe data are frequently noisy, and sensors may occasionally malfunction entirely. For a soft sensor to be resistant to such failure when predicting in real time, its underlying predictive model must be robust to outliers or corrupt data and missing data. Because of the difficult nature of predicting the process quality measurements such as the concentration of ethanol, a significant challenge in this modelling process is the scarcity of data. Experts in the field of process engineering [2] propose innovative solutions to the lack of data for modeling such as integrating first-principles knowledge (principles of physics and chemistry/biology) with data-driven models to develop hybrid models. Alternatively, we hypothesize that synthetically increasing the dataset using reliable data augmentation techniques will help to alleviate these issues. In a recent study, Demir et al. [3] affirm that the generation of synthetic data increases the robustness of the regression model to overfitting and boosts its accuracy.

The use of autoencoders and adversarial networks to augment data for regression problems has been studied in a growing number of contexts in the literature. Demir et al. [3] proposed the use of autoencoders (AEs), variational autoencoders (VAEs), and generative adversarial networks (GANs) to aid in expanding multivariate time series data on electricity demand forecasting for a regressive model. They found that while simple approaches such as gaussian jittering and scaling produced little benefit, the AEs, VAEs, and a variant of traditional GANs called a Wasserstein GAN [4] were able to reduce the mean absolute error by 2–3%. They noted that VAE and GAN architectures produced the best and most reliable results. Research by Papadopulos and Karalis [5] demonstrated the use of VAEs to augment data from clinical studies in contexts where increasing the number of participants has high cost and ethical considerations. They reported that the VAE data improved the statistical power of their analyses, particularly noting utility in datasets with a high level of variance. In the development of soft sensor technology, novel deep VAEs were used to solve missing data problems in an industrial polymerization process to produce polyamide/polyester [6]. Also, Shen et al. [7] proposed a multiresolution pyramid VAE in soft sensor modeling to solve the problem of the discrepant sampling rates of instruments in CO_2_ absorption columns in an industrial NH_3_ synthesis process. Wang and Liu [8] combined a VAE and a Wasserstein GAN to generate synthetic samples for the development of soft sensors to predict the rotor deformation of a power plant boiler air preheater.

This work takes our previous research [1] as a case study in fermentation processes and aims to leverage novel deep learning architectures with synthetic dataset generation to expand the size of the training dataset to ultimately improve the inferential power of the underlying model of a soft sensor. Specifically, we seek to discover whether generating synthetic datasets increases the robustness and predictive power of a regression model for predicting the concentration of ethanol during a fermentation process. Our methodology consists of three primary stages. In the first stage, we trained a VAE generative model on the original datasets (experimental measurements) to generate synthetic datasets. In the second stage, we trained FNN (Feedforward neural network) regression models constituting the soft sensor with identical architectures, one on original datasets (original model), one on a 50/50 blend of original plus synthetic datasets (Augmented10), and one on the blend of original plus 100 synthetic datasets (Augmented100). In the final stage, we tested all three models on original time series datasets (testing datasets) to evaluate their performance. The overall scheme of the research is shown in Figure 1.

## 2. Methods

### 2.1. Intensified Fermentation Process

Experimental datasets from several cycles of very-high-gravity (VHG) ethanol fermentation from sugarcane [9,10] are used to develop the soft sensor. The main objective of VHG fermentation, an intensified fermentation technology, was to maintain active and viable cells throughout the harvest season to provide a high rate of sugar conversion to ethanol. Eleven fermentation cycles were performed in a 3 L Bioflo 115 bioreactor (New Brunswick, NJ, USA) with a working volume of 2 L. Each cycle considered the conceptual process of the VHG ethanol fermentation [11]. A cascade of a decreasing 2 °C of temperature setpoint was applied according to the ethanol concentration which meant the highest temperature (34 °C) at the beginning and the lowest (28 °C) at the end of fermentation. Higher temperatures at the beginning of fermentation were considered to maximize conversion, and lower temperatures at the end of fermentation were considered to minimize inhibition and cell damage due to the high concentration of ethanol. At the last 3.5 h of fermentation, when it reached 108 g/L of ethanol, 0.2 vvm of air was sparged. Micro-aeration supports cell maintenance and the preservation of cellular activity. A two-step cell treatment was performed before each new cycle of fermentation. The first step is acid- washing using sulfuric acid followed by intracellular detoxification through centrifugation. The second step is a cell reactivation stage to promote membrane recovery and enzymatic restoration. Online probes of temperature, oxidation-reduction potential (redox potential), capacitance, and pH were used for monitoring the process. Measurements were taken at 1 min intervals. First and last samples of the concentration of ethanol were analyzed by high-performance liquid chromatography (HPLC) (1260, Agilent Technologies, Santa Clara, CA, USA) using an Aminex HPX-87P column (300 × 7.8 mm) and detected by a refractive index detector. Measurements of the remaining samples were conducted using mid-infrared spectroscopy. Samples were taken at 0, 2, 3, 5, 6, 8, and 10 h. These process analytical tools provided critical input–output data for developing the soft sensor. More details about the experimental setup and analytical procedures are described elsewhere [1].

### 2.2. Synthetic Data Generation

We began with the original dataset, which consists of eleven individual experiments, each containing about 500 timesteps, spanning around ten hours. Each experiment contained several variables, pH, redox potential, capacitance, and temperature, which comprise the four regression inputs, and the concentration of ethanol, the regression output. One experiment had a failed redox potential sensor, resulting in no data for that variable in the entire experiment, and this was imputed using a K-nearest neighbors (KNN) imputation technique via the scikit-learn Python library. In addition, another experiment had a deviating capacitance sensor that was likely uncalibrated and gave readings that were consistently far from that of all the other experiments. We will address the discussion of this noisy capacitance in the Results and Discussion section.

Using the open-source Time Series Generative Modelling (TSGM) library [12] built on the TensorFlow framework for Python, running in a Google Colab environment, we created a variational autoencoder (VAE) model. Having trained the model, we generated 100 synthetic experiments.

A VAE is a modification of the standard autoencoder (AE) that allows for the generation of new data. An AE consists of two components, an encoder and decoder. The encoder compresses a high-dimensionality space into a latent space, while the decoder performs a reconstruction into the original vector space [13]. In a VAE, random sampling is introduced to the decoding process such that new rather than identical data are generated. In this case, points are randomly sampled from the latent space then fed to the decoder.

The encoder yields the posterior distribution q(z|x), or the conditional probability of latent variable z given the evidence or real data x. In other words, the encoder produces a probable point in the latent space based on the real data. The decoder does the opposite, yielding the likelihood distribution p(x|z), or the conditional probability of some data x given the latent variable z as illustrated in Figure 1.

The objective function of the VAE is known as the evidence lower bound (ELBO), which the training process seeks to maximize [14]. The ELBO represents the lower bound of the log-likelihood function of the data. It consists of two terms. First is the reconstruction term, which is the log-likelihood of the reconstructed data at the decoder, expressed as an expectation. The second is the regularization term, which helps to prevent overfitting. It is given by the Kullback–Leibler (KL) divergence, an asymmetric measure of distance between distributions. In this case, the KL divergence is taken between the posterior distribution q(z|x) and the assumed prior distribution p(z). The standard normal is typically used for the prior distribution.

The formula for the ELBO is shown in Equation (1). The reconstruction term is the first term shown here. The KL divergence is the second term, and for the objective function it is subtracted since smaller KL divergence values are desirable.
(1)$$ELBO=E\left[\log p\left(x|z\right)\right]-D_{KL}\left(q\left(z|x\right)|p\left(z\right)\right)$$

The VAE architecture is as follows. In each epoch, the VAE encoder starts with the input layer of the time series data formatted as [500 × 7]. Each experiment includes 500 data points with 4 inputs (pH, redox, capacitance and temperature) and 3 outputs (ethanol, substrate, and cell concentrations), generating input and output data simultaneously. For simplicity, this study only presents the ethanol output. In the encoder, the input layer passes through five convolutional layers with a dropout layer, resulting in 500 × 64 feature maps. Each convolution layer used ReLU (Rectified Linear Unit) as an activation function. This is followed by a flattening layer, producing a 32,000-dimensional vector, which then passes through two dense layers with 512 and 64 units, respectively. The encoder then generates a latent layer with a mean and variance, both of size 10, and performs sampling to create a latent vector of size 10. The decoder then reverses this process to generate the output in the [500 × 7] format. The reconstruction loss and KL loss are calculated between the input and synthetic data. Multiple epochs are required to train the VAE.

### 2.3. FNN Training

We then developed an architecture for our regression model, opting for a simple feedforward neural network (FNN) [15] that was found to work well in preliminary testing. The model used a 4-80-60-1 (input–neurons in the first hidden layer–neurons in the second hidden layer–output) architecture. Consequently, the input training data were formatted as n sets × 500 data points × 4 variables, and the output training data were formatted as n sets × 500 data points × 1 variable. The n sets were stacked before being fed into the model.

When designing machine learning models, datasets are typically divided into training and testing sets; however, this process is not as straightforward because our data consists of eleven distinct time series. Because there is substantial variability between experiments, the model evaluation results were very sensitive to the way the data were split. To alleviate this problem, we settled on a repeated sampling method. In each iteration, we split the data by experiments, keeping entire time series grouped rather than mixing individual data points together. We then trained the model on the training dataset, then tested the models, recording the R^2^ and the root mean square error (RMSE) scores defined in Equations (2) and (3). This was repeated 100 times, and we then analyzed the resulting distribution of the scores.

To train the Original model, we employed the Leave Out One Cross Validation (LOOCV) technique. Of the 11 experiments, 10 were randomly selected to comprise the training datasets (10 sets × 500 data points), then the one remaining experiment was designated as the testing dataset. To train the Augmented10 model, we combined both approaches by randomly selecting 10 experiments from the original datasets and randomly selecting another 10 experiments from the synthetic datasets (total 20 sets × 500 data points) and choosing the remaining experiment from the original to be the testing dataset. To train the Augmented100 model, we used 10 original datasets and 100 synthetic datasets for training (total 110 sets × 500 data points) with the remaining original experiment serving as the testing dataset.

### 2.4. Test Metrics

At the completion of each iteration, we calculated the R^2^ and the RMSE scores [16] for the model based on the predictions generated for the concentration of ethanol from the original experiment in the testing dataset.

Note that the R^2^ score is the generalized coefficient of determination, which permits negative values. The interpretation is the ratio of variance explained by the regression model to the total variation in the data, where higher values are better up to a maximum of 1. Its formula is given by Equation (2).
(2)$$R^{2}\left(y,\hat{y}\right)=1-\frac{\sum_{i=1}^{n}{\left(y_{i}-{\hat{y}}_{i}\right)}^{2}}{\sum_{i=1}^{n}{\left(y_{i}-{\bar{y}}_{i}\right)}^{2}}$$
where y is the vector of true ethanol concentrations from the testing dataset, $\hat{y}$ is the vector of predictions from the model, $\bar{y}$ is the mean of the true ethanol values, and n is 500 timesteps.

An adjusted R^2^ provides a more accurate measure of a model’s explanatory power by accounting for the number of predictors relative to the number of observations as described by Equation (2). In our case, the number of predictors, *p*, remains the same before and after the augmented data.
(3)$${Adjusted\;R}^{2}\left(y,\hat{y}\right)=1-\frac{\left(1-R^{2})(N-1\right)}{N-p-1}$$

It should be noted that we used *N* as the number of sets of data rather than actual data points to account for stability and generalization ability with the increasing sets of data. So, instead of comparing with R^2^, as no additional predictors were added but the number of datasets increased, compare this metric between the original, Augmented10 and Augmented100.

MAE, the mean absolute error, is a measure of errors between paired observations expressing the same phenomenon. It is commonly used to evaluate the accuracy of predictive models. Its formula is given by Equation (4).
(4)$$MAE\left(y,\hat{y}\right)=\frac{1}{n}\sum_{i=1}^{n}\left(y_{i}-{\hat{y}}_{i}\right)$$

RMSE, or root mean squared error, is an aggregate measure of the errors between real and predicted values. Lowered values are better down to a minimum of 0. Its formula is given by Equation (5).
(5)$$RMSE\left(y,\hat{y}\right)=\sqrt{\frac{1}{n}\sum_{i=1}^{n}{\left(y_{i}-{\hat{y}}_{i}\right)}^{2}}$$

We then identified changes in the distributions from the original to the synthetic model, particularly noting the change in mean R^2^ score as the key indicator of change in predictive power. Secondly, we used the change in standard deviation of R^2^ scores as the indicator of change in the robustness since a narrower spread of the distribution represents a more consistent model.

## 3. Results

In this study, we trained a VAE generative model on the original datasets (experimental measurements) to generate synthetic data. We also initially consider the generation of generative adversarial network (GAN) datasets for comparison purposes. We used the TimeGAN [17] variant of the GAN that is designed for temporal data. Nonetheless, looking at the datasets generated by the GAN, it appears that it is not correctly accounting for the time series data and is producing unsatisfactory results (as seen in the bottom row in Figure 2). Wang and Liu [8] propose another, more sophisticated variant known as the Wasserstein GAN (WGAN); however, it was not implemented in the TSGM library used in this work. The application of variants of the GAN goes beyond the objectives of this study; therefore, we focused on the VAE deep learning approach that showed promising results.

Figure 2 displays the time series plots for each variable for each of the data sources (the original, VAE, and GAN). In the top row, for the original data, there are 11 curves corresponding to each of the 11 experiments. In the lower two rows, for the VAE and GAN datasets for comparison, there are 100 curves for the virtual experiments generated. We note that visually, the resemblance in distribution and temporality is strongest between the original datasets and the VAE datasets. The VAE-generated data follows a discernible temporal trend, effectively filling the dispersed gaps between experiments. In contrast, the GAN-generated data exhibit random fluctuations within the data spread. During the training of the VAE, the reconstruction loss decreased with the number of training epochs. Figure 3 shows the loss metrics for reconstruction and KL divergence over a training run of 250 epochs. Note that lower loss values are desirable and should be minimized [12]. Both loss metrics decrease up to about 100 epochs, after which they become irregular or increase. Consequently, we selected 100 epochs for our data generation process. Figure 4 displays the histograms for each variable from the original data, VAE-generated data, and GAN-generated data, both before and after data augmentation. It is evident that the VAE preserves the original data’s variability structure, as the histograms of the features before and after augmentation overlap well. In contrast, the GAN-generated data shows a distribution that deviates from the original datasets.

In the time series plot for capacitance (labeled Capacitance—original in Figure 2), there is one deviating curve that we have previously mentioned. Although a logical and tempting approach would be to exclude this experiment, sensors in practice are frequently noisy. One of the motivations of the work is to create a soft sensor robust against sensors feeding bad data, which therefore requires the inclusion of the experiment in our analysis. Surprisingly, the VAE handles the capacitance variable by creating a time series that fills in the spaces between the cluster of 10 experiments and the 1 outlier experiment.

This appears to be consistent with the t-distributed stochastic neighbor embedding (t-SNE) plot [18] with synthetic data superimposed on original data (Figure 5). t-SNE visualizes both the original and generated high-dimensional data in two dimensions for enhanced interpretability of data similarity. Figure 5a,b shows the t-SNE plots for the original and the generated datasets. The original datasets are marked with red triangles and the generated datasets are marked with blue hexagons for the VAE and green hexagons for the GAN. The observations indicate that datasets generated by the VAE not only cover the space occupied by the original data but also expand beyond it, forming a cohesive cluster. This suggests their capability to produce a novel dataset surpassing the confines of the existing dataset. Conversely, data generated by the GAN demonstrate a departure from the original data boundary, dispersed into adjacent areas, indicating a deficiency in accurately capturing the distribution of the original data. Based on these findings, we decided to use the VAE-generated data for the regression model stage.

The impact of the augmented dataset was tested using FNN regression models (Original, Augmented10, and Augmented100 models defined above), evaluated with R^2^ and RMSE scores. The statistics for the distribution of R^2^, adjusted R^2^, MAE, and RMSE scores from 100 runs of the regression model with the specified datasets are shown in Table 1. The %Δ columns display the change in the specified statistic using the original model as a baseline. For the Δ mean columns, we calculate the *p*-value using Welch’s unequal variance *t*-test of means [19]. For the Δ standard deviation columns, we calculate the *p*-value using the Levene test of equal variance [20].

The prediction of the original model resulted in a mean R^2^ and RMSE skewed by severe outliers visible in the histograms shown in Figure 6a,b. This indicates a high spread of the distribution and raises concerns about inconsistent model performance. In contrast, both the standard deviation of R^2^ and RMSE were greatly reduced and thus improve the consistency of the sensor in Augmented10 and Augmented100 shown in Figure 6c–f. We can also confirm this visually from the histograms in Figure 6, with R^2^ values clustering closer to 1 without the presence of extreme outliers.

Table 1 indicates that the introduction of synthetic datasets improves R^2^ means by 23% and 33% for Augmented10 and Augmented100 models, respectively. More dramatic improvements were observed in standard deviation. Predictions of Augmented10 and Augmented100 models show R^2^ values clustering closer to one without the presence of extreme outliers. The standard deviation for R^2^ was reduced by 79% and 82% for Augmented10 and Augmented100 models, respectively, with statistical significance at a *p* value less than 0.05. Adjusted R^2^ exhibited a more pronounced trend in line with the conventional R^2^, indicating that the augmented data are effectively enhancing the training set. This suggests that the added data improve the model’s ability to generalize, making the model more robust and reliable.

The improvements in MAE and RMSE were relatively moderate for the Augmented10 model, but the Augmented100 model showed 21%, a 24% reduction, respectively. The improvement in the standard deviation was more dramatic, showing a decrease of 57% and 63% for Augmented10 and Augmented100, respectively, for MAE, and 61% and 63% for RMSE. It is noted that for both metrics, the Augmented100 model yielded stronger improvements in both mean values and consistency. These results suggest that the underlying model of the soft sensor model could be more robust and consistent with the addition of synthetic datasets, leading to a more reliable output.

To illustrate the practical use of the soft sensor, we consider two cases: Anomalous data (represented by Experiment 9) and Baseline data (represented by Experiment 2) as testing datasets. The predicted ethanol concentration profiles by the soft sensor for these testing datasets are shown as orange lines in Figure 7. Experimental measurements of ethanol concentration are shown as blue markers for comparison. The Anomalous data (Experiment 9) features deviated capacitance data, whereas the Baseline data (Experiment 2) is arbitrarily selected from the original datasets. The results of the predictions using Anomalous data as the testing dataset are shown in panels (a), (c), and (e) for the Original, Augmented10, and Augmented100 models, respectively. Similarly, the results of the predictions using Baseline data as the testing dataset are shown in panels (b), (d), and (f) for the Original, Augmented10, and Augmented100 models, respectively.

Figure 7a,c,f demonstrates that the Augmented10 model substantially improved the predictive capability for the testing dataset in Anomalous data, but it remains insufficient. Remarkably, the Augmented100 model achieved satisfactory results by significantly enhancing the predictive capability. In Figure 7b,d,f, the results for Baseline data show sufficient prediction with the original dataset, with slight improvements observed using the Augmented10 and Augmented100 models.

Table 2 presents a more rigorous evaluation of the testing datasets using R² and RMSE scores. When Anomalous data were used as the testing dataset, the RMSE for predicting ethanol concentration improved dramatically with the Augmented10 and Augmented100 models, from −2.34 to 0.76 and 0.96, respectively, corresponding to improvements of 132% and 141%. RMSE values were reduced from 0.33 to 0.09 and 0.03, representing improvements of 72% and 91%, respectively. Similar satisfactory results, albeit to varying degrees, were observed when Baseline data were used as the testing dataset. The R² value for Baseline data increased from 0.88 to 0.90 and 0.97 with the Augmented10 and Augmented100 models.

## 4. Discussion

An essential aspect of developing reliable machine learning models is ensuring that the model generalizes well to unseen data, avoiding overfitting. In this study, we employed Leave-One-Out Cross-Validation (LOOCV) to rigorously assess the model’s performance. LOOCV is particularly advantageous as it uses each data instance exactly once as the test set while the remaining instances form the training set, providing a comprehensive evaluation of the model.

During the LOOCV process, we initially observed significant fluctuations in the model’s performance metrics when using the original dataset (Figure 6 and Table 1). These fluctuations indicated that the model was potentially overfitting to specific instances in the training data, thus failing to generalize effectively. To mitigate this issue, we augmented the dataset, which introduced more variability and helped the model to better capture the underlying patterns without overfitting to the noise. The improved scores with the synthetic data are likely a result of features inherent in the original data being incorporated into the variational autoencoder (VAE) training and propagating into the VAE synthetic data. However, in some iterations, the model, without being trained on a more comprehensive dataset, fails to account for the features of the original data. Specifically, the Augmented10 dataset performed less efficiently than the Augmented100 dataset because the sampled data were insufficient for effective regression model training. Thus, without a larger training dataset, the model cannot accurately account for the original data’s features.

We observed that, in general, the VAE-based strategy proposed in this study is useful in the development of soft sensors for fermentation processes. Training datasets with a well-distributed solution space are required for the effective learning of the fermentation process reaction rates, even in the presence of the non-homogeneity of the medium and probe malfunctions. Post-augmentation, the consistency and robustness of the soft sensor significantly improved, as evidenced by the more stable performance metrics across all iterations of the LOOCV. This improvement demonstrates that the augmented dataset facilitated better generalization of the model, thereby reducing the risk of overfitting. These findings emphasize the importance of rigorous validation techniques and data augmentation in developing robust machine learning models for complex processes such as fermentation.

Data-driven models are typically trained on specific datasets and may have limitations in their applicability in various operational scenarios. Moreover, data-driven modeling often lacks interpretability and has drawbacks in understanding and explaining the underlying mechanisms of the fermentation process. However, in our soft sensor application, the predictability in the presence of noise is the most important, demonstrating that such models can still be powerful tools for modeling fermentation processes. This methodology can also be applied to other bioprocesses.

The methodology applied in this study can be extended to other fermentation processes that utilize advanced online instrumentation such as infrared spectroscopy and fluorescence analysis, as well as conventional online measurements like off-gas analysis, pH, temperature, and dissolved oxygen. A data-driven modeling and synthetic time series can be used to predict important metrics like product titer or productivity, which are crucial for downstream processing or production indices, though such efforts often encounter challenges such as data scarcity and occasional measurement errors. Applying effective data augmentation techniques to time series and regression problems—typical in bioprocesses—can be challenging for researchers. This study illustrates practical strategies using the open-source TSGM Python library to generate time series datasets that can be applied to other bioprocesses, thereby enhancing the robustness and predictive power of soft sensors.

## 5. Conclusions

The method based on data-driven modeling and synthetic time series generation developed in this work provided a large number of synthetic datasets from eleven experimental datasets obtained from an intensified ethanol fermentation process. This is a significantly high performance generated from a small dataset of experimental measurements. Both the experimental (including outliers) and synthetic datasets were used to train the underlying predictive model of the soft sensor to predict the concentration of ethanol, a process quality measurement. By using the new dataset, the predictive power of the soft sensor to estimate the concentration of ethanol was increased by 34% and its variability was reduced by 82%, as shown by the mean and standard deviation of the R^2^ scores. Also, the soft sensor was able to be robust to outlier data of capacitance input signals. The developed method can provide a significant time and cost savings for data generation for modeling purposes either based on deep learning or mechanistic approaches. This work focused on the application of the method in an ethanol fermentation process, but its use in the study of other fermentation processes is straightforward.

## Figures and Tables

**Figure 1 bioengineering-11-00803-f001:**
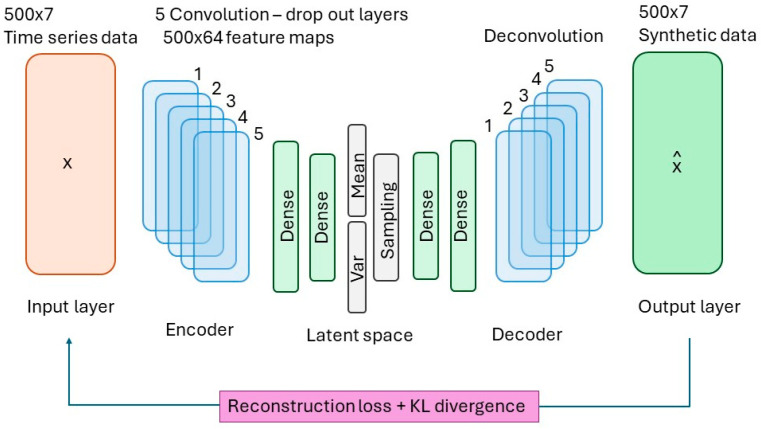
Convolutional variational autoencoder architecture. The deep learning network starts with the input data and then successively feeds into five convolutional layers. The output from the final convolutional layer is fed to the fully connected (dense) layers, which constructs the latent space. To reconstruct the contact maps, we use five successive de-convolutional layers, mirroring the forward convolutional layers.

**Figure 2 bioengineering-11-00803-f002:**
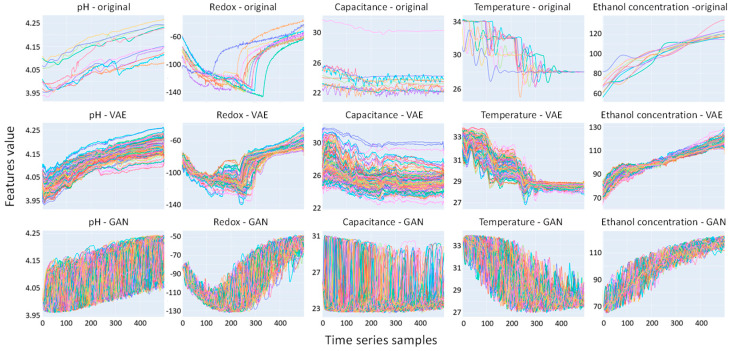
Time series for inputs: pH, redox (mV), capacitance (pF/cm), temperature (°C), and output: concentration of ethanol (g/L). The top row shows the time series of the original data from 11 experiments, the middle row shows the 100 VAE-generated datasets, and the bottom row shows the 100 GAN-generated datasets. The data shows the temporal progression of input and output data alongside the data spread.

**Figure 3 bioengineering-11-00803-f003:**
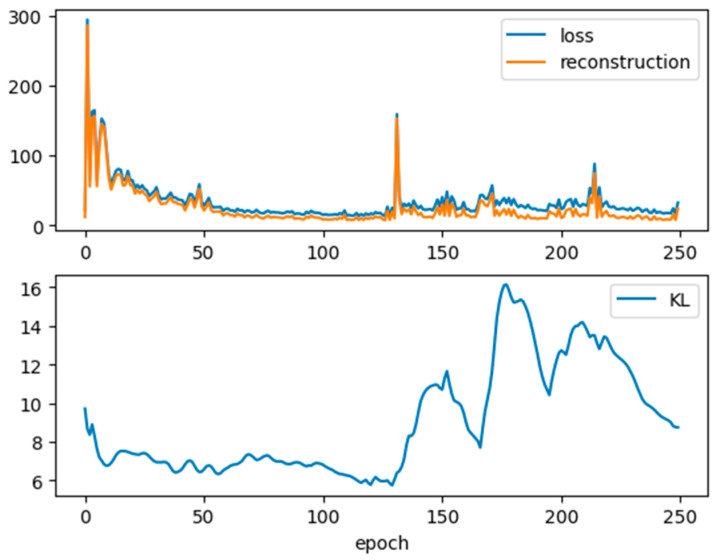
VAE training loss metrics for 250 epochs.

**Figure 4 bioengineering-11-00803-f004:**
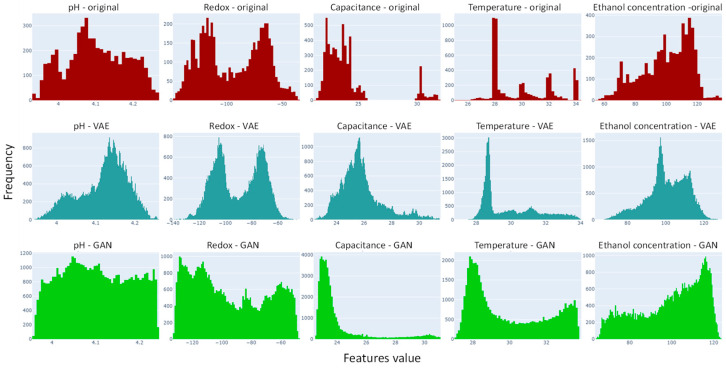
Histograms of features (inputs), pH, redox (mV), capacitance (pF/cm), temperature (°C), and output: concentration of ethanol (g/L) for data sources (original, VAE, and GAN).

**Figure 5 bioengineering-11-00803-f005:**
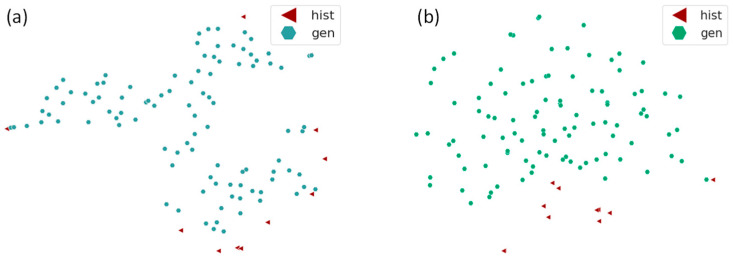
t-SNE plots for (**a**) the VAE and original data, and (**b**) for the GAN and original data. The original data are marked with red triangles and the generated data are marked with blue hexagons for the VAE and green hexagons for the GAN.

**Figure 6 bioengineering-11-00803-f006:**
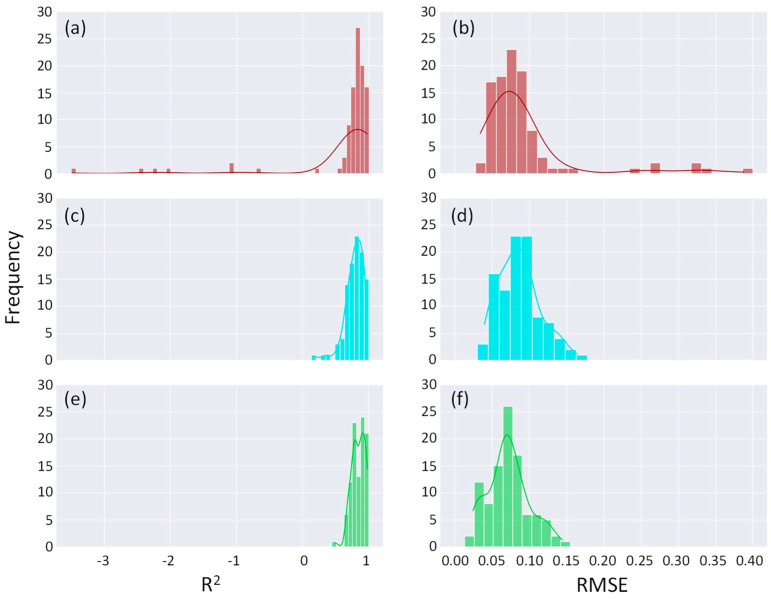
Histogram for R^2^ for (**a**) Original model, (**c**) Augmented10 model, and (**e**) Augmented100 model. Histogram for RMSE for (**b**) Original model, (**d**) Augmented10 model, and (**f**) Augmented100 model.

**Figure 7 bioengineering-11-00803-f007:**
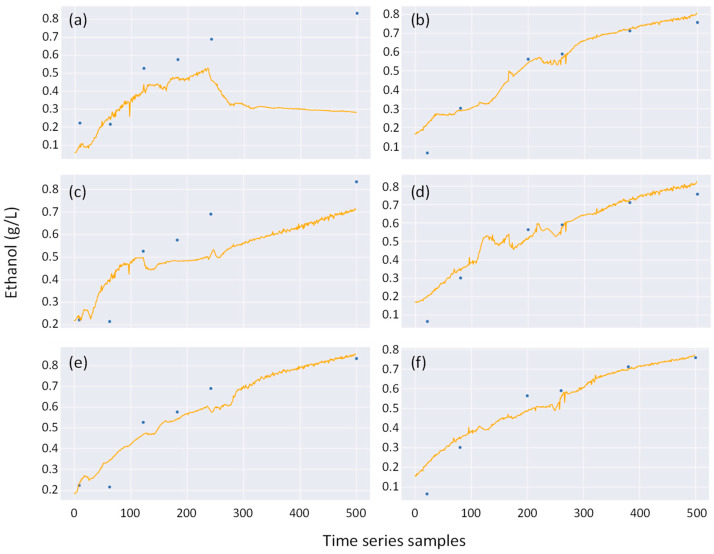
Predicted vs. original ethanol concentration for Anomalous data (represented by experiments 9) and Baseline data (represented by Exp. 2). The orange line represents the predicted ethanol concentration, while the blue markers represent the originally measured data, both with normalized values. The column of panels on the left shows the prediction results for Anomalous data using the (**a**) Original model, (**c**) Augmented10 model, and (**e**) Augmented100 model. The column of panels on the right shows the prediction results for Baseline data using the (**b**) Original model, (**d**) Augmented10 model, and (**f**) Augmented100 model.

**Table 1 bioengineering-11-00803-t001:** Distribution of R^2^ scores, Adjusted R^2^, MAE, and RMSE for 100 runs using the Original model, the Augmented10 model, and the Augmented100 model. Statistical significance is denoted by an asterisk (*) with the criteria of *p* < 0.05. (+) For the Adjusted R^2^, we used the number of datasets as N.

Model	Mean	%Δ Mean (*p*-Value)	Std. Dev.	%Δ Std. Dev. (*p*-Value)
R^2^				
Original	0.6367	–	0.7367	–
Augmented10	0.7872	+23.63% (*p* = 0.0470) *	0.1363	−79.46% (*p* = 0.040) *
Augmented100	0.8441	+33.98% (*p* = 0.006) *	0.097	−81.50% (*p* = 0.027) *
Adjusted R^2 (+)^				
Original	0.3461	–		
Augmented10	0.7304	+111.1% (*p* = 0.0001) *		
Augmented100	0.8381	+142.4% (*p* = 0.000) *		
MAE				
Original	0.0798	–	0.05387	–
Augmented10	0.0696	−12.7% (*p* = 0.210)	0.0253	−53.04% (*p* = 0.034) *
Augmented100	0.0582	−20.7% (*p* = 0.008) *	0.0206	−61.76% (*p* = 0.006) *
RMSE				
Original	0.09219	–	0.06449	–
Augmented10	0.08612	−6.58% (*p* = 0.3880)	0.02747	−57.40% (*p* = 0.067) *
Augmented100	0.06974	−24.35% (*p* = 0.0179) *	0.02369	−63.26% (*p* = 0.049) *

**Table 2 bioengineering-11-00803-t002:** R^2^ and RMSE calculated from the analysis of the comparison between the original (measurements) and predicted values of the concentration of ethanol for the training and testing datasets. Three regression models (Original, Augmented10, and Augmented100) developed were tested using Anomalous data (Experiments 9) and Baseline data (Experiment 2).

Model	Set	Prediction Performance of Concentration of Ethanol (g/L)			
		Case 1: Anomalous Data		Case 2: Baseline Data	
		R^2^	RMSE	R^2^	RMSE
Original	Training	0.95	0.4527	0.96	0.04168
	Testing	−2.37	0.3326	0.88	0.06499
Augmented10	Training	0.97	0.02289	0.97	0.02312
	Testing	0.76	0.09039	0.90	0.05629
Augmented100	Training	0.97	0.02337	0.97	0.02297
	Testing	0.96	0.02948	0.97	0.02499

## Data Availability

The data presented in this study are available on request from the corresponding author.

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
