# Peer review of "Enhancing Fermentation Process Monitoring through Data-Driven Modeling and Synthetic Time Series Generation"

_bioengineering, 2024, doi:10.3390/bioengineering11080803_

Round 1

Reviewer 1 Report

Comments and Suggestions for Authors

This work presents a study on the indirect monitoring of an ethanol fermentation process using a time series of experimental and synthetic measurements. The analysis tool corresponds to neural network regression models that combine the experimental and synthetic data sets. The results presented are interesting and show the potential of the method used. However, there are different aspects that are not described and should be clarified.

a) What operating conditions do the available experimental data present? Was an experimental design proposed to define the conditions? If so, what were the criteria? Clearly describe why the experimental data are representative of the ethanol production process, covering different operating conditions of the process.

 b) It is suggested to include general descriptions of how the synthetic time series are generated, the criteria for their generation, and how the training is performed. The basic fundamental theory of the methodology used, and the criteria used for managing time series are essential to understanding how the proposal was applied.

 c) The results only show two case studies, with little depth in the other cases' results. It is suggested that a Table that summarizes the results and discusses the scope of these results be included.

 d) It would be convenient to include more robust statistical methods to show the validation of the results. The R2 statistic is a fundamental metric.

e) The authors mention that their proposal can be extended to other fermentation processes, but they do not give details about it.

Comments on the Quality of English Language

None comments

Reviewer 2 Report

Comments and Suggestions for Authors

The paper "Enhancing fermentation process monitoring through data-driven modeling and synthetic time series generation" proposes a strategy to generate synthetic training data via data-driven methods to improve the modeling and prediction of fermentation processes. This paper is quite interesting and relevant in the field of bioprocess where it is quite hard to measure and estimate process variables. Generally, the paper is properly written, and easy to read and understand. Here are some suggestions to improve the quality of the paper.

1 Nowadays, convolutional neural networks within the Deep Learning framework are often used in bioprocesses, however, autoencoders are scarcely applied. Then it would be valuable to include a separate section (background) describing data-driven methods especially the fundamentals of autoencoders. It is suggested to move part of the subsection 2.1 generation stage to a new section data-driven background after the introduction.

2 In the discussion section, can the results be compared to similar approaches reported in the scientific literature to increase data in bioprocess? 

3 Does The proposed data-driven method have any drawbacks or limitation?

Reviewer 3 Report

Comments and Suggestions for Authors

The study aimed to improve the effectiveness of the soft sensor in a fermentation prediction process by creating synthetic datasets for training. I have some important suggestions.

1) The network used in the train phase can be explained with a figure.

2) The conclusion and discussion sections should be separated.

3) A figure showing general application will increase understandability.

4) "Testing Stage" title is not suitable for R2 and RMSE introduction and formulas. More metrics can be used and introduced in a separate section.

5) Figure 2 is not very clear, a box plot can be used instead.

6) The number of samples before and after the augmentation can be shown.

7) There should be a section that gives a brief introduction about the sensor and dataset used.

8) Has the overfitting status of the network been examined at the end of the training? So it should be added to the discussion section.

Round 2

Reviewer 1 Report

Comments and Suggestions for Authors

I consider that the observations made in the previous version of the manuscript have been addressed, so I consider that the work can now be published.

Author Response

Comment: I consider that the observations made in the previous version of the manuscript have been addressed, so I consider that the work can now be published.

Response: Thank you for your time to review the manuscript. 

Reviewer 3 Report

Comments and Suggestions for Authors

1) In the revised article, the 4th heading will be DISCUSSION.

2) The authors paid little attention to comment 5 and comment 7. These comments will facilitate the readability and comprehension of the article.

3) The graphic in comment 3 is not included in the article.
